# Lifelong Learning by Adjusting Priors

## Abstract

In representational lifelong learning an agent aims to learn to solve novel tasks while updating its representation in light of previous tasks. Under the assumption that future tasks are 'related' to previous tasks, representations should be learned in such a way that they capture the common structure across learned tasks, while allowing the learner sufficient flexibility to adapt to novel aspects of a new task. We develop a framework for lifelong learning in deep neural networks that is based on generalization bounds, developed within the PAC-Bayes framework. Learning takes place through the construction of a distribution over networks based on the tasks seen so far, and its utilization for learning a new task. Thus, prior knowledge is incorporated through setting a history-dependent prior for novel tasks. We develop a gradient-based algorithm implementing these ideas, based on minimizing an objective function motivated by generalization bounds, and demonstrate its effectiveness through numerical examples.

## 1 Introduction

Learning from examples is the process of inferring a general rule from a finite set of examples. It is well known in statistics (e.g., Devroye et al. (1996)) that learning cannot take place without prior assumptions. This idea has led in Machine Learning to the notion of inductive bias (Mitchell, 1980). Recent work in deep neural networks has achieved significant success in using prior knowledge in the implementation of structural constraints, e.g. the use of convolutions and weight sharing as building blocks, capturing the translational invariance of image classification. However, in general the relevant prior information for a given task is not always clear, and there is a need for building prior knowledge through learning from previous interactions with the world.

Learning from previous experience can take several forms: **Continual learning** - a single model is trained to solve a task which changes over time (and hopefully not 'forget' the knowledge from previous times, (e.g., Kirkpatrick et al. (2017)). **Multi-task learning** - the goal is to learn how to solve several observed tasks, while exploiting their shared structure. **Domain adaptation** - the goal is to solve a 'target' learning task using a single 'source' learning task (both are observed, but usually the target has mainly unlabeled data). **Lifelong Learning / Meta-Learning / Learning-to-Learn** - the goal is to extract knowledge from several observed tasks to be used for future learning on new (not yet observed) learning tasks. In contrast to multi-task learning, the performance is evaluated on the new tasks.

We work within the framework of *lifelong learning*, where an agent learns through interacting with the world, transferring the knowledge acquired along its path to any new task it encounters. This notion has been formulated by Baxter (2000) in a clear and simple context of 'task-environment'. In analogy to the standard single-task learning in which data is sampled from an unknown distribution, Baxter suggested to model a lifelong learning setting as if tasks are sampled from an unknown task distribution (environment), so that knowledge acquired from previous tasks can be used in order to improve performance on a novel task. Baxter's work not only provided an interesting and mathematically precise perspective for lifelong learning, but also provided generalization bounds demonstrating the potential improvement in performance due to prior knowledge. Baxter's seminal work, has led to a large number of extensions and developments.

In this contribution we work within the framework formulated by Baxter (2000), and, following the setup in Pentina & Lampert (2014), provide generalization error bounds within the PAC-Bayes framework. These bounds are then used to develop a practical learning algorithm that is applied to

neural networks, demonstrating the utility of the approach. The main contributions of this work are the following. *(i)* An improved and tighter bound in the theoretical framework of Pentina & Lampert (2014) which can utilize different single-task PAC-Bayesian bounds. *(ii)* Developing a learning algorithm within this general framework and its implementation using probabilistic feedforward neural networks. This yields transfer of knowledge between tasks through constraining the prior distribution on a learning network. *(iii)* Empirical demonstration of the performance enhancement compared to naive approaches and recent methods in this field.

As noted above, Baxter (2000) provided a basic mathematical formulation and initial results for life-long learning. While there have been many developments in this field since then (e.g., Andrychow-icz et al. (2016); Edwards & Storkey (2016); Finn et al. (2017); Ravi & Larochelle (2016)), most of them were not based on generalization error bounds which is the focus of the present work. An elegant extension of generalization error bounds to lifelong learning was provided by Pentina & Lampert (2014), mentioned above (more recently extended in Pentina & Lampert (2015)). Their work, however, did not provide a practical algorithm applicable to deep neural networks. More recently, Dziugaite & Roy (2017) developed a single-task algorithm based on PAC-Bayes bounds that was demonstrated to yield good performance in simple classification tasks. Other recent theoretical approaches to lifelong or multitask learning (e.g. Alquier et al. (2017); Maurer et al. (2016)) provide increasingly general bounds but have not led directly to practical learning algorithms.

## 2 BACKGROUND: PAC-BAYES LEARNING

In the standard setting for supervised learning a set of (usually) independent pairs of input/output samples $S = \{(x_i, y_i)\}_{i=1}^m$ are given, each sample drawn from an *unknown* probability distribution $D$, namely $(x_i, y_i) \sim D$. We will use the notation $S \sim D^m$ to denote the distribution over the full sample. The usual learning goal is, based on $S$ to find a function $h \in \mathcal{H}$, where $\mathcal{H}$ is the so-called hypothesis space, that minimizes the expected loss function $\mathbb{E}\ell(h, z)$, where $z = (x, y)$ [1] and $\ell(h, z)$ is a *loss function* bounded in $[0, 1]$. As the distribution $D$ is unknown, learning consists of selecting an appropriate $h$ based on the sample $S$. In classification $\mathcal{H}$ is a space of classifiers mapping the input space to a finite set of classes. As noted in the Introduction, an inductive bias is required for effective learning. While in the standard approach to learning, described in the previous paragraph, one usually selects a single classifier (e.g., the one minimizing the empirical error), the PAC-Bayes framework, first formulated by McAllester (1999), considers the construction of a complete probability distribution over $\mathcal{H}$, and the selection of a single hypothesis $h \in \mathcal{H}$ based on this distribution. Since this distribution depends on the data it is referred to as a *posterior distribution* and will be denoted by $Q$. We note that while the term 'posterior' has a Bayesian connotation, the framework is not necessarily Bayesian, and the posterior does not need to be related to the prior through the likelihood function as in standard Bayesian analysis. The PAC-Bayes framework has been widely studied in recent years, and has given rise to significant flexibility in learning, and, more importantly, to some of the best generalization bounds available Audibert (2010); McAllester (2013); Lever et al. (2013). The framework has been recently extended to the lifelong learning setting by Pentina & Lampert (2014), and will be extended and applied to neural networks in the present contribution.

### 2.1 SINGLE-TASK PROBLEM FORMULATION

Following the notation introduced above we define the generalization error and the empirical error used in the standard learning setting,

$$er(h, D) \triangleq \mathop{\mathbb{E}}_{z \sim D} \ell(h, z) \quad ; \quad \widehat{er}(h, S) \triangleq \frac{1}{m} \sum_{j=1}^m \ell(h, z_i) \quad (h \in \mathcal{H}) . \tag{1}$$

Since the distribution $D$ is unknown, $er(h, D)$ cannot be directly computed.

---

[1] Note that the framework is not limited to supervised learning and can also handle unsupervised learning.

**PAC-Bayesian learning**    In the PAC-Bayesian setting the learner outputs a distribution over the entire hypothesis space $\mathcal{H}$, i.e, the goal is to provide a *posterior* distribution $Q \in \mathcal{M}$, where $\mathcal{M}$ denotes the set of distributions over $\mathcal{H}$. The expected (over $\mathcal{H}$) *generalization error* and *empirical error* are then given in this setting by averaging (1) over the posterior distribution,

$$er\left(Q, D\right) \triangleq \underset{h \sim Q}{\mathbb{E}} \underset{z \sim D}{\mathbb{E}} \ell\left(h, z\right) \quad ; \quad \widehat{er}\left(Q, S\right) \triangleq \underset{h \sim Q}{\mathbb{E}} \frac{1}{m} \sum_{j=1}^{m} \ell\left(h, z_j\right) \quad (Q \in \mathcal{M}). \tag{2}$$

This average describes a *Gibbs prediction* procedure - first drawing a hypothesis $h$ from $Q$ then applying it on the sample $z$. Similarly to (1), the generalization error $er\left(Q, D\right)$ cannot be directly computed since $D$ is unknown.

## 2.2    PAC-BAYESIAN GENERALIZATION BOUND

In this section we introduce a PAC-Bayesian bound for the single-task setting. The bound will also serve us for the lifelong-learning setting in the next sections. PAC-Bayesian bounds are based on specifying some reference distribution $P \in \mathcal{M}$. $P$ is called the *'prior'* since it must not depend on the observed data $S$. The distribution over hypotheses $Q$ which is provided as an output from the learning process is called the posterior (since it is allowed to depended on $S$). [2]

The classical PAC-Bayes theorem for single-task learning was formulated by McAllester (1999).

**Theorem 1** (McAllester's single-task bound)**.** *Let $P \in \mathcal{M}$ be some prior distribution over $\mathcal{H}$. Then for any $\delta \in (0, 1]$,*

$$\mathbb{P}_{S \sim D^m} \left\{ er\left(Q, D\right) \leq \widehat{er}\left(Q, S\right) + \sqrt{\frac{1}{2(m-1)} \left(D_{KL}(Q||P) + \log \frac{m}{\delta}\right)}, \forall Q \in \mathcal{M} \right\} \geq 1 - \delta \,, \tag{3}$$

where $D_{KL}(Q||P)$ is the Kullback-Leibler Divergence (KLD),

$$D_{KL}(Q||P) \triangleq \underset{h \sim Q}{\mathbb{E}} \log \frac{Q(h)}{P(h)}. \tag{4}$$

The bound (3) can be interpreted as stating that with high probability the expected error $er\left(Q, D\right)$ is upper bounded by the empirical error plus a complexity term. Since, with high probability, the bound holds for all $Q \in \mathcal{M}$ (uniformly), we can choose $Q$ after observing the data $S$. By choosing $Q$ that minimizes the bound we will get a learning algorithm with generalization guarantees. Note that PAC-Bayesian bounds express a trade-off between fitting the data (empirical error) and a complexity/regularization term (distance from prior) which encourages selecting a 'simple' hypothesis, namely one similar to the prior. The contribution of the prior-dependent regularization term to the objective is more significant for a smaller data set. For asymptotically large sample size $m$, the complexity term converges to zero. The specific choice of $P$ affects the bound's tightness and so should express prior knowledge about the problem. Generally, we want the prior to be close to posteriors which can achieve low training error. For example, we may want to use a prior that prefers simpler hypotheses (Occam's razor).

In general, the bound might not be tight, and can even be vacuous, i.e., greater than the maximal value of the loss. However, Dziugaite & Roy (2017) recently showed that a PAC-Bayesian bound can achieve non-vacuous values with deep-networks and real data sets. In this work our focus is on deriving an algorithm for lifelong-learning, rather than on actual calculation of the bound. We expect that even if the numerical value of the bound is vacuous, it still captures the behavior of the generalization error and so minimizing the bound is a good learning strategy.

In our experiments we will plug in positive and unbounded loss functions in the bound, in contrast to the assumption on a bounded loss. Theoretically, we can still claim that we are bounding a variation of the loss clipped to $[0, 1]$. Furthermore, empirically the loss function are almost always smaller than one.

---

[2]As noted above, the terms 'prior' and 'posterior' might be misleading, since, this is not a Bayesian inference setting (the prior an posterior are not connected trough the Bayes rule). However, PAC-Bayes and Bayesian analysis have interesting and practical connections, as we will see in the next sections (see also Germain et al. (2016)).

## 3 PAC-BAYESIAN LIFELONG-LEARNING

In this section we introduce the lifelong-learning setting. In this setting a lifelong-learning agent observes several 'training' tasks from the same task environment. The lifelong-learner must extract some common knowledge from these tasks, which will be used for learning new tasks from the same environment. In the literature this setting is often called learning-to-learn, meta-learning or lifelong-learning. We will formulate the problem and provide a generalization bound which will later lead to a practical algorithm. Our work extends Pentina & Lampert (2014) and establishes a more general bound. Furthermore, we will demonstrate how to apply this result practically in non-linear deep models using stochastic learning.

### 3.1 LIFELONG LEARNING PROBLEM FORMULATION

The lifelong learning problem formulation follows Pentina & Lampert (2014). We assume all tasks share the sample space $\mathcal{Z}$, hypothesis space $\mathcal{H}$ and loss function $\ell : \mathcal{H} \times \mathcal{Z} \to [0, 1]$. The learning tasks differ in the unknown sample distribution $D_t$ associated with each task $t$. The lifelong-learning agent observes the training sets $S_1, ..., S_n$ corresponding to $n$ different tasks. The number of samples in task $i$ is denoted by $m_i$. Each observed dataset $S_i$ is assumed to be generated from an unknown sample distribution $S_i \sim D_i^{m_i}$. As in Baxter (2000), we assume that the sample distributions $D_i$ are generated $i.i.d.$ from an unknown tasks distribution $\tau$.

The goal of the lifelong-learner is to extract some knowledge from the observed tasks that will be used as prior knowledge for learning new (yet unobserved) tasks from $\tau$. The prior knowledge comes in the form of a distribution over hypotheses, $P \in \mathcal{M}$. When learning a new task, the learner uses the observed task's data $S$ and the prior $P$ to output a posterior distribution $Q(S, P)$ over $\mathcal{H}$. We assume that all tasks are learned via the same learning process. Namely, for a given $S$ and $P$ there is a specific output $Q(S, P)$. Hence $Q()$ is a function: $Q : \mathcal{Z}^m \times \mathcal{M} \to \mathcal{M}$. [3]

The quality of a prior $P$ is measured by the expected loss when using it to learn new tasks, as defined by,

$$er(P, \tau) \triangleq \underset{(D,m) \sim \tau}{\mathbb{E}} \underset{S \sim D^m}{\mathbb{E}} \underset{h \sim Q(S,P)}{\mathbb{E}} \underset{z \sim D}{\mathbb{E}} \ell(h, z). \tag{5}$$

Since we want to prove a PAC-Bayes style bound for lifelong-learning, we assume that the lifelong-learner does not select a single prior $P$, but instead infers a distribution $\mathcal{Q}$ over all prior distributions in $\mathcal{M}$. [4] Since $\mathcal{Q}$ is inferred *after* observing the tasks, it is called the hyper-posterior distribution, and serves as a prior for a new task, i.e., when learning a new task, the learner draws a prior from $\mathcal{Q}$ and then uses it for learning.

Ideally, the performance of the hyper-posterior $\mathcal{Q}$ is measured by the expected generalization loss of learning new tasks using priors generated from $\mathcal{Q}$. This quantity is denoted as the *transfer error*

$$er(\mathcal{Q}, \tau) \triangleq \underset{P \sim \mathcal{Q}}{\mathbb{E}} er(P, \tau). \tag{6}$$

While $er(\mathcal{Q}, \tau)$ is not computable, we can however evaluate the *empirical multi-task error*

$$\widehat{er}(\mathcal{Q}, S_1, ..., S_n) \triangleq \underset{P \sim \mathcal{Q}}{\mathbb{E}} \frac{1}{n} \sum_{i=1}^{n} \widehat{er}(Q(S_i, P), S_i). \tag{7}$$

Although $er(\mathcal{Q}, \tau)$ cannot be evaluated, we will prove a PAC-Bayes style upper bound on it, that can be minimized over $\mathcal{Q}$. It is important to emphasize that the hyper-posterior is evaluated on new, independent, tasks from the environment (and not on the observed tasks which ae used for meta-training).

In the single-task PAC-Bayes setting one selects a prior $P \in \mathcal{M}$ before seeing the data, and updates it to a posterior $Q \in \mathcal{M}$ after observing the training data. In the present lifelong setup, following the framework in Pentina & Lampert (2014), one selects an initial hyper-prior distribution $\mathcal{P}$, essentially

---

[3]In the next section we will use stochastic optimization methods as learning algorithms, but we can assume convergence to a same solution for any execution with a given $S$ and $P$.

[4]After proving the bound, we will simplify the lifelong-learning objective into choosing a single optimal prior.

a distribution over prior distributions $P$, and, following the observation of the data from all tasks, updates it to a hyper-posterior distribution $\mathcal{Q}$. As a simple example, assume the initial prior $P$ is a Gaussian distribution over neural network weights, characterized by a mean and covariance. A hyper distribution would correspond in this case to a distribution over the mean and covariance of $P$.

## 3.2 LIFELONG-LEARNING PAC-BAYESIAN BOUND

In this section we present a novel bound on the transfer error in the lifelong learning setup. The theorem is proved in the appendix 8.1.

**Theorem 2** (Lifelong-learning PAC-Bayes bound). *Let $Q : \mathcal{Z}^m \times \mathcal{M} \to \mathcal{M}$ be a mapping (single-task learning procedure), and let $\mathcal{P}$ be some predefined hyper-prior distribution. Then for any $\delta \in (0, 1]$ the following inequality holds uniformly for all hyper-posteriors distributions $\mathcal{Q}$ with probability of at least $1 - \delta$,* [5]

$$er(\mathcal{Q}, \tau) \leq \frac{1}{n} \sum_{i=1}^{n} \mathop{\mathbb{E}}_{P \sim \mathcal{Q}} \widehat{er}_i(Q_i(S_i, P), S_i) + \tag{8}$$

$$\frac{1}{n} \sum_{i=1}^{n} \sqrt{\frac{1}{2(m_i - 1)} \left( D_{KL}(\mathcal{Q}||\mathcal{P}) + \mathop{\mathbb{E}}_{P \sim \mathcal{Q}} D_{KL}(Q(S_i, P)||P) + \log \frac{2nm_i}{\delta} \right)} +$$

$$\sqrt{\frac{1}{2(n - 1)} \left( D_{KL}(\mathcal{Q}||\mathcal{P}) + \log \frac{2n}{\delta} \right)}.$$

Notice that the transfer error (6) is bounded by the empirical multi-task error (7) plus two complexity terms. The first is the average of the task-complexity terms of the observed tasks. This term converges to zero in the limit of a large number of samples in each task ($m_i \to \infty$). The second is an environment-complexity term. This term converges to zero if infinite number of tasks is observed from the task environment ($n \to \infty$). As in Pentina & Lampert (2014), our proof is based on two main steps. The second step, similarly to Pentina & Lampert (2014), bounds the generalization error at the task-environment level (i.e, the error caused by observing only a finite number of tasks), $er(\mathcal{Q}, \tau)$, by the average generalization error in the observed tasks plus the environment-complexity term.

The first step differs from Pentina & Lampert (2014). Instead of using a single joint bound on the average generalization error, we use a single-task PAC-Bayes theorem to bound the generalization error in each task separately (when learned using priors from the hyper-posterior), and then use a union bound argument. By doing so our bound takes into account the specific number of samples in each observed task (instead of their harmonic mean). Therefore our bound is better adjusted the observed data set.

Another distinction is in the case in which an infinite number of tasks is observed, but each has only a few samples. In contrast to Pentina & Lampert (2014), in Theorem 2 the hyper-prior still has an effect on the bound. Intuitively, the prior knowledge we had before observing tasks (hyper-prior) should still have an effect on the bound unless the observed tasks contain enough information (samples).

Our proof technique can utilize different single-task bounds in each of the two steps. In section 8.1 we use McAllester's bound (Theorem 1), which is tighter than the lemma used in Pentina & Lampert (2014). Therefore, the complexity terms are in the form of $\sqrt{\frac{1}{m} D_{KL}(Q||P)}$ instead of $\frac{1}{\sqrt{m}} D_{KL}(Q||P)$ as in Pentina & Lampert (2014). This means the bound is tighter (e.g. see Seldin et al. (2012) Theorems 5 and 6). In section 8.2 we demonstrate how our technique can use other, possibly tighter, single-task bounds. Finally, in the experiments section we empirically evaluate the transfer risk obtained when using the bounds as learning objectives and show that our bound leads to far better results.

---

[5] The probability is taken over sampling of $(D_i, m_i) \sim \tau$ and $S_i \sim D_i^{m_i}, i = 1, ..., n$.

# 4 LIFELONG-LEARNING ALGORITHM

As in the single-task case, the bound of Theorem 2 can be evaluated from the training data and so can serve as a minimization objective for a principled lifelong-learning algorithm. Since the bound holds uniformly for all $\mathcal{Q}$, it is ensured to hold also for the inferred optimal $\mathcal{Q}^*$. In this section we will derive a practical learning procedure that can applied to a large family of differentiable models, including deep neural networks.

## 4.1 HYPER-POSTERIOR MODEL

In this section we will choose a specific form for the Hyper-posterior distribution $\mathcal{Q}$, which enables practical implementation. Given a parametric family of priors $\left\{P_\theta : \theta \in \mathbb{R}^{N_P}\right\}$, the space of hyper-posteriors consists of all distributions over $\mathbb{R}^{N_P}$. We will limit our search to a certain family of hyper-posteriors by choosing a Gaussian distribution in the space of prior parameters,

$$\mathcal{Q}_{\theta_P} \triangleq \mathcal{N}\left(\theta_P, \kappa_{\mathcal{Q}}^2 I_{N_P \times N_P}\right), \tag{9}$$

where $\kappa_{\mathcal{Q}} > 0$ is a predefined constant.

Notice that $\mathcal{Q}$ appears in the bound (8) in two forms (i) divergence from the hyper-prior $D_{KL}(\mathcal{Q}||\mathcal{P})$ and (ii) expectations over $P \sim \mathcal{Q}$.

First, by setting the hyper-prior as Gaussian, $\mathcal{P} = \mathcal{N}\left(0, \kappa_{\mathcal{P}}^2 I_{N_P \times N_P}\right)$, where $\kappa_{\mathcal{P}} > 0$ is another constant, we get a simple form for the KLD term,

$$D_{KL}(\mathcal{Q}_{\theta_P}||\mathcal{P}) = \frac{1}{2\kappa_{\mathcal{P}}^2} \|\theta_P\|_2^2. \tag{10}$$

Note that the hyper-prior serves as a regularization term for learning the prior.

Second, the expectations can be easily approximated using by averaging several Monte-Carlo samples of $P$. Notice that sampling from $\mathcal{Q}_{\theta_P}$ means adding Gaussian noise to the prior parameters $\theta_P$ during training, $\theta'_P = \theta_P + \varepsilon_P, \varepsilon_P \sim \mathcal{N}\left(0, \kappa_{\mathcal{Q}}^2 I_{N_P \times N_P}\right)$. This means the learned parameters must be robust to perturbations, which encourages selecting solutions which are less prone to over-fitting and are expected to generalize better (Chaudhari et al., 2016; Keskar et al., 2016).

## 4.2 JOINT OPTIMIZATION

The term appearing on the RHS of the lifelong learning bound in (8) can be compactly written as

$$J(\theta_P) \triangleq \frac{1}{n} \sum_{i=1}^n J_i(\theta_P) + \Upsilon(\theta_P), \tag{11}$$

where we defined,

$$J_i(\theta_P) \triangleq \underset{P \sim \mathcal{Q}_{\theta_P}}{\mathbb{E}} \widehat{er}_i\left(Q_i(S_i, P), S_i\right) + \tag{12}$$

$$\sqrt{\frac{1}{2(m_i - 1)}\left(D_{KL}(\mathcal{Q}_{\theta_P}||\mathcal{P}) + \underset{P \sim \mathcal{Q}_{\theta_P}}{\mathbb{E}} D_{KL}(Q(S_i, P)||P) + \log \frac{2nm_i}{\delta}\right)},$$

and

$$\Upsilon(\theta_P) \triangleq \sqrt{\frac{1}{2(n-1)}\left(D_{KL}(\mathcal{Q}_{\theta_P}||\mathcal{P}) + \log \frac{2n}{\delta}\right)}. \tag{13}$$

Theorem 2 allows us to choose *any* single-task learning procedure $Q(S_i, P) : \mathcal{Z}^{m_i} \times \mathcal{M} \to \mathcal{M}$ to infer a posterior. We will use a procedure which minimizes $J_i(\theta_P)$ due to the following advantages: *(i)* It minimizes a bound on the generalization error of the observed task (see section 8.1). *(ii)* It uses the prior knowledge gained from the prior $P$ to get a tighter bound and a better learning objective. *(iii)* As will be shown next, formulating the single task learning as an optimization problem enables joint learning of the shared prior and the task posteriors.

To formulate the single-task learning as an optimization problem, we choose a parametric form for the posterior of each task $Q_{\phi_i}, \phi_i \in \mathbb{R}^{N_Q}$ (see section 4.3 for an explicit example). The single-task learning algorithm can be formulated as $\phi_i^* = \operatorname{argmin}_{\phi_i} J_i(\theta_P, \phi_i)$, where we abuse notation by denoting the term $J_i(\theta_P)$ evaluated with posterior parameters $\phi_i$ as $J_i(\theta_P, \phi_i)$.

The lifelong-learning problem of minimizing $J(\theta_P)$ over $\theta_P$ can now be written more explicitly,

$$\min_{\theta_P, \phi_1, \ldots, \phi_n} \left\{ \frac{1}{n} \sum_{i=1}^n J_i(\theta_P, \phi_i) + \Upsilon(\theta_P) \right\}. \tag{14}$$

### 4.3 DISTRIBUTIONS MODEL

In this section we make the lifelong-learning optimization problem (14) more explicit by defining a model for the posterior and prior distributions. First, we define the hypothesis class $\mathcal{H}$ as a family of functions parameterized by a *weight vector* $\{h_w : w \in \mathbb{R}^d\}$. Given this parameterization, the posterior and prior are distributions over $\mathbb{R}^d$.

We will present an algorithm for any differentiable model [6], but our aim is to use neural network (NN) architectures. In fact, we will use Stochastic NNs (Graves, 2011; Blundell et al., 2015) since in our setting the weights are random and we are optimizing their posterior distribution. The techniques presented next will be mostly based on Blundell et al. (2015).

Next we define the posteriors $Q_{\phi_i}$ and the prior $P_{\theta_P}$ as factorized Gaussian distributions [7],

$$P_{\theta_P}(w) = \prod_{k=1}^d \mathcal{N}\left(w_k; \mu_{P,k}, \sigma_{P,k}^2\right) \quad ; \quad Q_{\phi_i}(w) = \prod_{k=1}^d \mathcal{N}\left(w_k; \mu_{i,k}, \sigma_{i,k}^2\right), \ i = 1, \ldots, n, \tag{15}$$

where for each task, the posterior parameters vector $\phi_i = (\mu_i, \rho_i) \in \mathbb{R}^{2d}$ is composed of the means and log-variances of each weight , $\mu_{i,k}$ and $\rho_{i,k} = \log \sigma_{P,k}^2, k = 1, \ldots, d$.[8] The shared prior vector $\theta_P = (\mu_P, \rho_P) \in \mathbb{R}^{2d}$ has a similar structure. Since we aim to use deep models where $d$ could be in the order of millions, distributions with more parameters might be impractical.

Since $Q_{\phi_i}$ and $P_{\theta_P}$ are factorized Gaussian distributions the KLD takes a simple analytic form,

$$D_{KL}(Q_{\phi_i} || P_{\theta_P}) = \frac{1}{2} \sum_{k=1}^d \left\{ \log \frac{\sigma_{P,k}^2}{\sigma_{i,k}^2} + \frac{\sigma_{i,k}^2 + (\mu_{i,k} - \mu_{P,k})^2}{\sigma_{P,k}^2} - 1 \right\}. \tag{16}$$

### 4.4 OPTIMIZATION TECHNIQUE

As an underlying optimization method, we will use stochastic gradient descent (SGD)[9]. In each iteration, the algorithm takes a parameter step in a direction of an estimated negative gradient. As is well known, lower variance facilitates convergence and its speed. Recall that each single-task bound is composed of an empirical error term and a complexity term (12). The complexity term is a simple function of $D_{KL}(Q_{\phi_i} || P_{\theta_P})$ (16), which can easily be differentiated analytically. However, evaluating the gradient of the empirical error term is more challenging.

Recall the definition of the empirical error, $\widehat{er}(Q_{\phi_i}, S_i) = \mathbb{E}_{w \sim Q_{\phi_i}}(1/m_i) \sum_{j=1}^{m_i} \ell(h_w, z_{i,j})$. This term poses two major challenges. *(i)* The data set $S_i$ could be very large making it expensive to cycle over all the $m_i$ samples. *(ii)* The term $\ell(h_w, z_j)$ might be highly non-linear in $w$, rendering the expectation intractable. Still, we can get an unbiased and low variance estimate of the gradient.

First, instead of using all of the data for each gradient estimation we will use a randomly sampled mini-batch $S_i' \subset S_i$. Next, we require an estimate of a gradient of the form $\nabla_\phi \mathbb{E}_{w \sim Q_\phi} f(w)$ which

---

[6]The only assumption on $\{h_w : w \in \mathbb{R}^d\}$ is that the loss function $\ell(h_w, z)$ is differentiable w.r.t $w$.

[7]This choice makes optimization easier, but in principle we can use other distributions as long as the PDF is differentiable w.r.t the parameters.

[8]Note that we use $\rho = \log \sigma^2$ as a parameter in order to keep the parameters unconstrained (while $\sigma^2 = \exp(\rho)$ is guaranteed to be strictly positive).

[9]Or some other variant of SGD.

is a common problem in machine learning. We will use the 'reparametrization trick' (Rezende et al., 2014; Kingma & Welling, 2013) which is an efficient and low variance method [10] . The re-parametrization trick is easily applicable in our setup since we are using Gaussian distributions. The trick is based on describing the Gaussian distribution $w \sim Q_{\phi_i}$ (15) as first drawing $\varepsilon \sim \mathcal{N}(\bar{0}, I_{d \times d})$ and then applying the deterministic function $w(\phi_i, \varepsilon) = \mu_i + \sigma_i \odot \varepsilon$ (where $\odot$ is an element-wise multiplication).

Therefore, we can switch the order of gradient and expectation to get

$$\nabla_\phi \mathop{\mathbb{E}}_{w \sim Q_\phi} f(w) = \nabla_\phi \mathop{\mathbb{E}}_{\varepsilon \sim \mathcal{N}(\bar{0}, I_{d \times d})} f(w(\phi_i, \varepsilon)) = \mathop{\mathbb{E}}_{\varepsilon \sim \mathcal{N}(\bar{0}, I_{d \times d})} \nabla_\phi f(w(\phi_i, \varepsilon)).$$

The expectation can be approximated by averaging a small number of Monte-Carlo samples with reasonable accuracy. For a fixed sampled $\varepsilon$, the gradient $\nabla_\phi f(w(\phi_i, \varepsilon))$ is easily computable with backpropagation.

In summary, the Lifelong learning by Adjusting Priors (LAP) algorithm is composed of two phases In the first phase (Algorithm 1, termed "meta-training") several observed "training tasks" are used to learn a prior. In the second phase (Algorithm 2, termed "meta-testing") the previously learned prior is used for the learning of a new task (which was unobserved in the first phase). Note that the first phase can be used independently as a multi-task learning method. Both algorithms are described in pseudo-code in the appendix (section 8.4).

## 5   TOY EXAMPLE ILLUSTRATION

To illustrate the setup visually, we will consider a simple toy example of a 2D estimation problem. In each task, the goal is to estimate the mean of the data generating distribution. In this setup, the samples $z$ are vectors in $\mathbb{R}^2$. The hypothesis class is a the set of 2D vectors, $h \in \mathbb{R}^2$. As a loss function we will use the Euclidean distance, $\ell(h, z) \triangleq \|h - z\|_2^2$. We artificially create the data of each task by generating 50 samples from the appropriate distribution: $\mathcal{N}\left((2, 1)^\top, 0.1^2 I_{2 \times 2}\right)$ in task 1, and $\mathcal{N}\left((4, 1)^\top, 0.1^2 I_{2 \times 2}\right)$ in task 2. The prior and posteriors are 2D factorized Gaussian distributions, $P \triangleq \mathcal{N}\left(\mu_P, \mathrm{diag}(\sigma_P^2)\right)$ and $Q_i \triangleq \mathcal{N}\left(\mu_i, \mathrm{diag}(\sigma_i^2)\right), i = 1, 2$.

We run Algorithm 1 (meta-training) with complexity terms according to Theorem 1. As seen in Figure 1, the learned prior (namely, the prior learned from the two tasks) and single-task posteriors can be understood intuitively. First, the posteriors are located close to the ground truth means of each task, with relatively small uncertainty covariance. Second, the learned prior is located in the middle between the two posteriors, and its covariance is larger in the first dimension. This is intuitively reasonable since the prior learned that tasks are likely to have values of around 1 in dimension 2 and values around 3 in the dimension 1, but with larger variance. Thus, new similar tasks can be learned using this prior with fewer samples.

## 6   EXPERIMENTAL RESULTS

In this section we demonstrate the performance of our transfer method with image classification tasks solved by deep neural networks. [11].

In image classification, the data samples, $z \triangleq (x, y)$, are pairs of a an image, $x$, and a label, $y$. The hypothesis class $\left\{h_w : w \in \mathbb{R}^d\right\}$ is a the set of neural networks with a given architecture (which will be specified later). As a loss function $\ell(h_w, z)$ we will use the cross-entropy loss.

We conduct an experiment with a task environment in which each task is created by a random permutation of the labels of the MNIST dataset (LeCun, 1998). The meta-training set is composed of 5 tasks from the environment, each with $60,000$ training examples. Following the meta-training

---

[10]In fact, we will use the '**local** re-parameterization trick' (Kingma et al., 2015) in which we sample a different $\varepsilon$ for each data point in the batch, which reduces the variance of the estimate. To make the computation more efficient with neural-networks, the random number generation is performed w.r.t the activations instead of the weights (see Kingma et al. (2015) for more details.).

[11]Our implementation uses the PyTorch framework. The code for reproducing all the experiments can be found in the GitHub repository: https://github.com/ML-Paper/lifelong-learning-pt.

phase, the learned prior is used to learn a new meta-test task with fewer training samples $(2,000)$. The network architecture is a small CNN with 2 convolutional-layers, a linear hidden layer and a linear output layer. See section 8.3 for more implementation details.

We compare the average generalization performance (test error) in learning a new (meta-test) when using the following methods.

As a baseline, we measure the performance of learning without transfer from the training-tasks:

Figure 1: **Toy example:** the orange are red dots are the samples of task 1 and 2, respectively, and the green and purple dots are the means of the posteriors of task 1 and 2, respectively. The mean of the prior is a blue dot. The ellipse around each distribution's mean represents the covariance matrix.

- **Scratch-standard**: standard learning from scratch (non-stochastic network).
- **Scratch-stochastic**: stochastic learning from scratch (stochastic network with no prior/complexity term).

Other methods transfer knowledge from only one of the train tasks:

- **Warm-start-transfer**: Standard learning with initial weights taken from the standard learning of a single task from the meta-train set (with $60,000$ examples).
- **Oracle-transfer**: Same as the previous method, but all layers besides the output layer are frozen (unchanged from their initial value), which is a common practice for transfer learning in computer vision (Razavian et al., 2014). Note that in this method we are manually inserting prior knowledge based on our familiarity with the task environment. Therefore this method can be considered an "oracle".

Finally, we compare methods which transfer knowledge from all of the training tasks:

- **LAP-M**: The objective is based on Theorem 2 - the lifelong-learning bound obtained using Theorem 3 (McAllester's single-task bound).
- **LAP-S**: The objective is based on the lifelong-learning bound of eq. (23) in section 8.2. This lifelong-learning bound is obtained using Theorem 4 (Seeger's single-task bound).
- **LAP-PL**: In this method we use the main theorem of Pentina & Lampert (2014) as an objective for the algorithm, instead of Theorem 2.
- **LAP-KLD**: Here we use a task-complexity term which is simply the KLD between the sampled prior and the task posterior and an environment-complexity term which is the KLD between the hyper-prior and hyper-posterior. This minimization problem is equivalent to maximization of the Evidence-Lower-Bound (ELBO) when using a variational methods to approximate the maximum-likelihood parameters of a hierarchical generative model [12]. Note that the ELBO can also be interpreted as an upper bound on the generalization error. However the bound is looser than the one obtained using PAC-Bayesian methods [13].
- **Averaged-prior**: Each of the training tasks is learned in a standard way to obtain a weights vector, $w_i$. The learned prior is set as an isotropic Gaussian with unit variances and a mean vector which is the average of $w_i, i = 1, .., n$. This prior is used for meta-testing as in LAP-S.
- **MAML**: The Model-Agnostic-Meta-Learning (MAML) algorithm by Finn et al. (2017) finds an optimal initial weight for learning tasks from a given environment. We report the best results obtained with all combinations of the following representative hyper-parameters: 1-3 gradient steps in meta-training, 1-20 gradient steps in meta-testing and $\alpha \in \{0.01, 0.1, 0.4\}$.

---

[12]For example, see Edwards & Storkey (2016), equation 4.
[13]See discussion in section 3.2.

Table 1: Comparing the test error of different learning methods on 100 test tasks (average $\pm$ STD)

| Method | Average | STD |
|---|---|---|
| **Scratch-standard** | 2.27% | 0.16% |
| **Scratch-stochastic** | 2.8% | 0.16% |
| **Warm-start-transfer** | 1.43% | 0.12% |
| **Oracle-transfer** | 0.802% | 0.06% |
| **LAP-M** | 1.04% | 0.1% |
| **LAP-S** | 0.856% | 0.08% |
| **LAP-PL** | 61.3% | 15.9% |
| **LAP-KLD** | 91.0% | 6.06% |
| **Averaged-Prior** | 2.89% | 0.19% |
| **MAML** | 0.931% | 0.06% |

As can be seen in Table 1, the best results are obtained with the "oracle" method. Recall that the oracle method has the "unfair" advantage of a "hand-engineered" transfer technique which is based on knowledge about the problem. In contrast, the other methods must automatically learn the task environment by observing several tasks.

The LAP-M and LAP-S variants of the LAP algorithm improves considerably over learning from scratch and over the naive warm-start transfer and are very close the the "oracle" method. As expected the the LAP-S variant preforms better since it uses a tighter bound (see section 8.2).

The other variants of the LAP algorithm with other objectives performed much worse. First, the results for LAP-PL demonstrate the importance of the tight generalization bound developed in our work. Second, the results for the LAP-KLD show that deriving objectives from variational-inference techniques that maximize a lower bound on the model evidence, might be less a successful approach than deriving objectives which minimize upper bounds on the generalization error.

The results for the "averaged-prior" method are about the same as learning from scratch. Due to the high non-linearity of the problem, averaging weights was not expected to perform well .

The results of MAML are comparable to the results of the LAP algorithm. Note that MAML is specifically suited for learning from many few-shot tasks, in which taking a small number of gradient steps in each task is effective for learning. However, in our experiment, there are a few tasks but more than a few samples. Still, the method performed quite well with several different sets of hyperparameters [14].

## 7 DISCUSSION AND FUTURE WORK

We have presented a framework for representational lifelong learning, motivated by PAC-Bayes generalization bounds, and implemented through the adjustment of a learned prior, based on tasks encountered so far. The framework bears conceptual similarity to the empirical Bayes method while not being Bayesian, and is implemented at the level of tasks rather than samples. Combining the general approach with the rich representational structure of deep neural networks, and learning through gradient based methods leads to an efficient procedure for lifelong learning, as motivated theoretically and demonstrated empirically. While our experimental results are preliminary, we believe that our work attests to the utility of using rigorous performance bounds to derive learning algorithms, and demonstrates that tighter bounds indeed lead to improved performance.

There are several open issues to consider. First, the current version learns to solve all available tasks in parallel, while a more useful procedure should be sequential in nature. This can be easily incorporated into our framework by updating the prior following each novel task. Second, our method requires training stochastic models which is challenging due to the the high-variance gradients. We we would like to develop new methods within our framework which have more stable convergence and are easier to apply in larger scale problems. Third, there is much current effort in reinforcement

---

[14]The best results for MAML were obtained $\alpha = 0.01$, 2 gradient steps in meta-training and 16 in meta-testing.

learning to augment model free learning with model based components, where some aspects of the latter are often formulated as supervised learning tasks. Incorporating our approach in such a context would be a worthwhile challenge. In fact, a similar framework to ours was recently proposed within an RL setting (Teh et al., 2017), although it was not motivated from performance guarantees as was our approach, but rather from intuitive heuristic arguments.

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

## 8 APPENDIX

### 8.1 PROOF OF THE LIFELONG-LEARNING BOUND

In this section we prove Theorem 2. The proof is based on two steps, both use McAllaster's classical PAC-Bayes bound. In the first step we use it to bound the error which is caused due to observing only a finite number of samples in each of the observed tasks. In the second step we use it again to bound the generalization error due to observing a limited number of tasks from the environment.

We start by restating the classical PAC-Bayes bound (McAllester, 1999; Shalev-Shwartz & Ben-David, 2014) using general notations.

**Theorem 3** (Classical PAC-Bayes bound, general formulation). *Let $\mathcal{X}$ be some 'sample' space and $\mathbb{X}$ some distribution over $\mathcal{X}$. Let $\mathcal{F}$ be some 'hypothesis' space. Define a 'loss function' $g(f,X):\mathcal{F}\times\mathcal{X}\to[0,1]$. Let $X_1^K \triangleq \{X_1,...,X_K\}$ be a sequence of $K$ independent random variables distributed according to $\mathbb{X}$. Let $\pi$ be some prior distribution over $\mathcal{F}$ (which must not depend on the samples $X_1,...,X_K$). For any $\delta \in (0,1]$, the following bound holds uniformly for all 'posterior' distributions $\rho$ over $\mathcal{F}$ (even sample dependent),*

$$\mathbb{P}_{X_1^K \underset{i.i.d}{\sim} \mathbb{X}} \Bigg\{ \underset{X\sim\mathbb{X}}{\mathbb{E}} \underset{f\sim\rho}{\mathbb{E}} g(f,X) \le \frac{1}{K}\sum_{k=1}^{K} \underset{f\sim\rho}{\mathbb{E}} g(f,X_k) + \tag{17}$$

$$\sqrt{\frac{1}{2(K-1)}\left(D_{KL}(\rho||\pi)+\log\frac{K}{\delta}\right)}, \forall\rho \Bigg\} \ge 1-\delta.$$

**First step** We use Theorem 3 to bound the generalization error in each of the observed tasks when learning is done by an algorithm $Q:\mathcal{Z}^{m_i}\times\mathcal{M}\to\mathcal{M}$ which uses a prior and the samples to output a distribution over hypothesis.

Let $i\in 1,...,n$ be the index of some observed task. We use Theorem 3 with the following substitutions. The samples are $X_k\triangleq z_{i,j}$, $K\triangleq m_i$, and their distribution is $\mathbb{X}\triangleq D_i$. We define a 'tuple hypothesis' $f=(P,h)$ where $P\in\mathcal{M}$ and $h\in\mathcal{H}$. The 'loss function' is the regular loss which uses only the $h$ element in the tuple, $g(f,X)\triangleq\ell(h,z)$. We define the 'prior over hypothesis', $\pi\triangleq(\mathcal{P},P)$, as some distribution over $\mathcal{M}\times\mathcal{H}$ in which we first sample $P$ from $\mathcal{P}$ and then sample $h$ from $P$. According to Theorem 3, the 'posterior over hypothesis' can be any distribution (even sample dependent), in particular, the bound will hold for the following family of distributions over $\mathcal{M}\times\mathcal{H}$, $\rho\triangleq(\mathcal{Q},Q(S_i,P))$, in which we first sample $P$ from $\mathcal{Q}$ and then sample $h$ from $Q=Q(S_i,P)$. [15]

The KLD term is,

$$D_{KL}(\rho||\pi)=\underset{f\sim\rho}{\mathbb{E}}\log\frac{\rho(f)}{\pi(f)}=\underset{P\sim\mathcal{Q}}{\mathbb{E}}\underset{h\sim Q(S,P)}{\mathbb{E}}\log\frac{\mathcal{Q}(P)Q(S_i,P)(h)}{\mathcal{P}(P)P(h)}=$$

$$\underset{P\sim\mathcal{Q}}{\mathbb{E}}\log\frac{\mathcal{Q}(P)}{\mathcal{P}(P)}+\underset{P\sim\mathcal{Q}}{\mathbb{E}}\underset{h\sim Q(S,P)}{\mathbb{E}}\log\frac{Q(S_i,P)(h)}{P(h)}=$$

$$D_{KL}(\mathcal{Q}||\mathcal{P})+\underset{P\sim\mathcal{Q}}{\mathbb{E}}D_{KL}(Q(S_i,P)||P)$$

Plugging in to (17) we obtain that for any $\delta_i>0$

$$\mathbb{P}_{S_i\sim D_i^m}\Bigg\{\underset{z\sim D_i}{\mathbb{E}}\underset{P\sim\mathcal{Q}}{\mathbb{E}}\underset{h\sim Q(S_i,P)}{\mathbb{E}}\ell(h,z)\le\frac{1}{m_i}\sum_{j=1}^{m_i}\underset{P\sim\mathcal{Q}}{\mathbb{E}}\underset{h\sim Q(S_i,P)}{\mathbb{E}}\ell(h,z_{i,j})+ \tag{18}$$

$$\sqrt{\frac{1}{2(m_i-1)}\left(D_{KL}(\mathcal{Q}||\mathcal{P})+\underset{P\sim\mathcal{Q}}{\mathbb{E}}D_{KL}(Q(S_i,P)||P)+\log\frac{m_i}{\delta_i}\right)},\forall\mathcal{Q}\Bigg\}\ge 1-\delta_i,$$

for all observed tasks $i=1,..,n$.

---

[15]Recall that $Q(S_i,P)$ is the posterior distribution which is the output of the learning algorithm $Q()$ which uses the data $S_i$ and the prior $P$.

Using the terms in section 2.1, we can write the above as,

$$
\mathbb{P}_{S_i \sim D_i^m} \Bigg\{ \underset{P \sim \mathcal{Q}}{\mathbb{E}} \, er\left(Q(S_i, P), D_i\right) \leq \underset{P \sim \mathcal{Q}}{\mathbb{E}} \, \widehat{er}\left(Q(S_i, P), S_i\right) + \tag{19}
$$

$$
\sqrt{\frac{1}{2(m_i - 1)} \left( D_{KL}(\mathcal{Q}||\mathcal{P}) + \underset{P \sim \mathcal{Q}}{\mathbb{E}} D_{KL}(Q(S_i, P)||P) + \log\frac{m_i}{\delta_i} \right)}, \forall \mathcal{Q} \Bigg\} \geq 1 - \delta_i,
$$

**Second step** Next we wish to bound the environment-level generalization (i.e, the error due to observing only a finite number of tasks from the environment). We will use Theorem 3 again, with the following substitutions. The i.i.d samples are $(D_i, m_i, S_i), i = 1, ..., n$ where $(D_i, m_i)$ are distributed according to the task-distribution $\tau$ and $S_i \sim D_i^{m_i}$. The 'hypothesis' are $f \triangleq P$ and the 'loss function' is $g(f, X) \triangleq \underset{h \sim Q(S,P)}{\mathbb{E}} \underset{z \sim D}{\mathbb{E}} \ell(h, z)$. Let $\pi \triangleq \mathcal{P}$ be some distribution over $\mathcal{M}$, the bound will hold uniformly for all distributions $\rho \triangleq \mathcal{Q}$ over $\mathcal{M}$.

For any $\delta_0 > 0$, the following holds (according to Theorem 3),

$$
\mathbb{P}_{(D_i, m_i) \sim \tau, S_i \sim D_i^{m_i}, i=1,..,n} \Bigg\{ \underset{(D,m) \sim \tau}{\mathbb{E}} \underset{S \sim D^m}{\mathbb{E}} \underset{P \sim \mathcal{Q}}{\mathbb{E}} \underset{h \sim Q(S,P)}{\mathbb{E}} \underset{z \sim D}{\mathbb{E}} \ell(h, z) \leq \tag{20}
$$

$$
\frac{1}{n} \sum_{i=1}^{n} \underset{P \sim \mathcal{Q}}{\mathbb{E}} \underset{h \sim Q(S_i, P)}{\mathbb{E}} \underset{z \sim D_i}{\mathbb{E}} \ell(h, z) + \sqrt{\frac{1}{2(n-1)} \left( D_{KL}(\mathcal{Q}||\mathcal{P}) + \log\frac{n}{\delta_0} \right)}, \forall \mathcal{Q} \Bigg\} \geq 1 - \delta_0.
$$

Using the terms in section 3.1, we can write the above as,

$$
\mathbb{P}_{(D_i, m_i) \sim \tau, S_i \sim D_i^{m_i}, i=1,..,n} \Bigg\{ er\left(\mathcal{Q}, \tau\right) \leq \underset{P \sim \mathcal{Q}}{\mathbb{E}} \frac{1}{n} \sum_{i=1}^{n} er\left(Q(S_i, P), D_i\right) + \tag{21}
$$

$$
\sqrt{\frac{1}{2(n-1)} \left( D_{KL}(\mathcal{Q}||\mathcal{P}) + \log\frac{n}{\delta_0} \right)}, \forall \mathcal{Q} \Bigg\} \geq 1 - \delta_0.
$$

Finally, we will bound the probability of the event which is the intersection of the events in (19) and (21) by using the union bound. For any $\delta > 0$, set $\delta_0 \triangleq \frac{\delta}{2}$ and $\delta_i \triangleq \frac{\delta}{2n}$ for $i = 1, ..., n$.

Using the union bound we finally get,

$$
\mathbb{P}_{(D_i, m_i) \sim \tau, S_i \sim D_i^{m_i}, i=1,...,n} \Bigg\{ er\left(\mathcal{Q}, \tau\right) \leq \frac{1}{n} \sum_{i=1}^{n} \underset{P \sim \mathcal{Q}}{\mathbb{E}} \widehat{er}_i\left(Q_i(S_i, P), S_i\right) +
$$

$$
\frac{1}{n} \sum_{i=1}^{n} \sqrt{\frac{1}{2(m_i - 1)} \left( D_{KL}(\mathcal{Q}||\mathcal{P}) + \underset{P \sim \mathcal{Q}}{\mathbb{E}} D_{KL}(Q(S_i, P)||P) + \log\frac{2nm_i}{\delta} \right)} +
$$

$$
\sqrt{\frac{1}{2(n-1)} \left( D_{KL}(\mathcal{Q}||\mathcal{P}) + \log\frac{2n}{\delta} \right)}, \forall \mathcal{Q} \Bigg\} \geq 1 - \delta.
$$

## 8.2 LIFELONG LEARNING BOUND BASED ON ALTERNATIVE SINGLE-TASK BOUNDS

Many PAC-Bayesian bounds for single-task learning have appeared in the literature. In this section we demonstrate how our proof technique can be used with a different single-task bound to derive a possibly tighter lifelong-learning bound.

Consider the following single-task bound by (Seeger, 2002; Maurer, 2004). [16]

**Theorem 4** (Seeger's single-task bound). *Under the same notations as Theorem 3, for any $\delta \in (0, 1]$ we have,*

$$
\mathbb{P}_{X_1,...,X_K \underset{i.i.d}{\sim} \mathbb{X}} \Bigg\{ \underset{X \sim \mathbb{X}}{\mathbb{E}} \underset{f \sim \rho}{\mathbb{E}} g(f, X) \leq \widehat{er}\left(\rho, X_1^K\right) + 2\varepsilon + \sqrt{2\varepsilon\widehat{er}\left(\rho, X_1^K\right)}, \forall \rho \Bigg\} \geq 1 - \delta,
$$

---

[16]Note that we used the slightly tighter version version by Maurer (2004) bound which requires $K \geq 8$ .

*where we define,*

$$\varepsilon(K, \rho, \pi, \delta) = \frac{1}{K} \left( D_{KL}(\rho||\pi) + \log \frac{2\sqrt{K}}{\delta} \right),$$

*and,*

$$\widehat{er}\left(\rho, X_1^K\right) = \frac{1}{K} \sum_{k=1}^{K} \mathop{\mathbb{E}}_{f \sim \rho} g(f, X_k).$$

Using the above theorem we get an alternative intra-task bound to (19),

$$\mathbb{P}_{S_i \sim D_i^m} \Bigg\{ \mathop{\mathbb{E}}_{P \sim \mathcal{Q}} er\left(Q(S_i, P), D_i\right) \le \mathop{\mathbb{E}}_{P \sim \mathcal{Q}} \widehat{er}\left(Q(S_i, P), S_i\right) + \tag{22}$$

$$2\varepsilon_i + \sqrt{2\varepsilon_i \widehat{er}\left(Q(S_i, P), S_i\right)}, \forall \mathcal{Q} \Bigg\} \ge 1 - \delta_i,$$

where,

$$\varepsilon_i = \frac{1}{m_i} \left( D_{KL}(\mathcal{Q}||\mathcal{P}) + \mathop{\mathbb{E}}_{P \sim \mathcal{Q}} D_{KL}(Q(S_i, P)||P) + \log \frac{2\sqrt{m_i}}{\delta_i} \right).$$

While the classical bound of Theorem 1 converges at a rate of about $1/\sqrt{m}$ (as in basic VC-like bounds), the bound of Theorem 4 converges even faster (at a rate if $\frac{1}{m}$) if the empirical error $\widehat{er}(Q)$ is negligibly small (compared to $\frac{1}{m} D_{KL}(Q||P)$). Since this is commonly the case in modern deep learning, we expect this bound to be tighter than others in this regime. [17]

By utilizing the Theorem 4 in the first step of the proof in section 8.1 we can get a tighter bound for lifelong-learning:

$$\mathbb{P}_{(D_i, m_i) \sim \tau, S_i \sim D_i^{m_i}, i=1,\ldots,n} \Bigg\{ er\left(\mathcal{Q}, \tau\right) \le \frac{1}{n} \sum_{i=1}^{n} \left[ \mathop{\mathbb{E}}_{P \sim \mathcal{Q}} \widehat{er}_i\left(Q_i(S_i, P), S_i\right) + \tag{23} \right.$$

$$\left. 2\varepsilon_i + \sqrt{2\varepsilon_i \widehat{er}\left(Q(S_i, P), S_i\right)} \right] + \sqrt{\frac{1}{2(n-1)} \left( D_{KL}(\mathcal{Q}||\mathcal{P}) + \log \frac{2n}{\delta} \right)}, \forall \mathcal{Q} \Bigg\} \ge 1 - \delta,$$

where, $\varepsilon_i$ is defined in (22) (and $\delta_i = \frac{\delta}{2n}$).

### 8.3 CLASSIFICATION EXAMPLE IMPLEMENTATION DETAILS

The network architecture used for the permuted-labels experiment is a small CNN with 2 convolutional-layers of 10 and 20 filters, each with $5 \times 5$ kernels, a hidden linear layer with 50 units and a linear output layer. Each convolutional layer is followed by max pooling operation with kernel of size 2. Dropout with $p = 0.5$ is performed before the output layer. In both networks we use ELU (Clevert et al., 2015) (with $\alpha = 1$) as an activation function. Both phases of the LAP algorithm (algorithms 1 and 2) ran for 200 epochs, with batches of 128 samples in each task. We take only one Monte-Carlo sample of the stochastic network output in each step. As optimizer we used ADAM (Kingma & Ba, 2014) with learning rate of $10^{-3}$. The means of the weights ($\mu$ parameters) are initialized randomly by $\mathcal{N}\left(0, 0.1^2\right)$, while the log-var of the weights ($\rho$ parameters) are initialized by $\mathcal{N}\left(-10, 0.1^2\right)$. The hyper-prior and hyper-posterior parameters are $\kappa_{\mathcal{P}} = 2000$ and $\kappa_{\mathcal{Q}} = 0.001$ respectively and the confidence parameter was chosen to be $\delta = 0.1$ .

To evaluate the trained network we used the maximum of the posterior for inference (i.e. we use only the means the weights). [18]

---

[17] More recent works presented possibly tighter PAC-Bayesian bounds by taking into account the empirical variance (Tolstikhin & Seldin, 2013) or by specializing the bound deep for neural networks (Neyshabur et al., 2017). However, we leave the incorporation of these bounds for future work.

[18] Classifying using the the majority vote of several runs gave similar results in this experiment.

## 8.4 PSEUDO CODE

---

**Algorithm 1:** LAP algorithm, meta-training phase (learning-to-learn)

---

**Input** : Data sets of observed tasks: $S_1, ..., S_n$
**Output:** Learned prior parameters $\theta_P$

**Initialize:**

- $\theta_P = (\mu_P, \rho_P) \in \mathbb{R}^d \times \mathbb{R}^d$
- $\phi_i = (\mu_i, \rho_i) \in \mathbb{R}^d \times \mathbb{R}^d, \quad \text{for} \quad i = 1, ..., n$

**while** *not done* **do**

    **for** *each task $i \in \{1, ..n\}$* [19] **do**

        - Sample a random mini-batch from the data $S_i' \subset S_i$
        - Approximate $J_i(\theta_P, \phi_i)$ (12) using $S_i'$ and averaging Monte-Carlo draws

    **end**

    - $J \leftarrow \frac{1}{n} \sum_{i \in \{1, ..n\}} J_i(\theta_P, \phi_i) + \Upsilon(\theta_P)$
    - Evaluate the gradient of $J$ w.r.t $\{\theta_P, \phi_1, ..., \phi_n\}$ using backprop
    - Take an optimization step

**end**

---

**Algorithm 2:** LAP algorithm, meta-testing phase (learning a new task)

---

**Input** : Data set of a new task, $S$, and prior parameters, $\theta_P$
**Output:** Posterior parameters $\phi'$ which solve the new task

**Initialize:**

- $\phi' \leftarrow \theta_P$

**while** *not done* **do**

    - Sample a random mini-batch from the data $S' \subset S$
    - Approximate the empirical loss $J$ (12) using $S'$ and averaging Monte-Carlo draws
    - Evaluate the gradient of $J$ w.r.t $\phi'$ using backprop
    - Take an optimization step

**end**

---

[19]For implementation considerations, when training with a large number of tasks we can sample a subset of tasks in each iteration ("meta min-batch" ) to estimate $J$.

