# OpenReview forum: "Lifelong Learning by Adjusting Priors"
_ICLR.cc/2018/Conference — Reject_

### Official Review · AnonReviewer1 · 2017-11-20
**Well written paper, with very week experiments.**

**Rating:** 6
**Confidence:** 4

**Review:**

The author extends existing PAC-Bayes bounds to multi-task learning, to allow the prior to be adapted across different tasks. Inspired by the variational bayes literature, a probabilistic neural network is used to minimize the bound. Results are evaluated on a toy dataset and a synthetically modified version of MNIST.

While this paper is well written and addresses an important topic, there are a few points to be discussed:

* Experimental results are really week. The toy experiment only compares the mean of two gaussians. Also, on the synthetic MNIST experiments, no comparison is done with any external algorithms. Neural Statistician, Model-Agnostic Meta-Learning and matching networks all provide decent results on such setup. While it is tolerated to have minimal experiments in a theoretical papers, the theory only extends Pentina & Lampert (2014). Also, similar algorithms can be obtain through variational-bayes evidence lower bound.

* The bound appears to be sub-optimal. A bound where the KL term vanishes by 1/n would probably be tighter. I went in appendix to try to see how the proof could be adapted but it’s definitively not as well written as the rest of the paper. I’m not against putting proofs in appendix but only when it helps clarity. In this case it did not.

* The paper is really about multi-task learning. Lifelong learning implies some continual learning and addressing the catastrophic forgetting issues. I would recommend against overuse of the lifelong learning term.

Minors:
* Define KLD
* Section 5.1 : “to toy”
* Section 5.1.1: “the the”

---

> ### Author Response · Authors · 2017-12-25
> **We thank the reviewer for the helpful comments, and address the  specific points below.**
>
> * The toy example (section 5) was meant only for visualization of the setup.
> In the revised version we separated it from the experimental results section.
>
> In the experimental results part (section 6) we added a comparison to the recently introduced Model-Agnostic Meta-Learning (MAML, Finn, Abbeel, and Levine. "Model-Agnostic Meta-Learning for Fast Adaptation of Deep Networks." arXiv preprint arXiv:1703.03400 (2017).) .
>
> We also addressed the comparison to variational methods  which maximize the evidence lower bound (section 6).  Actually, such methods can be seen as minimizing a bound on the generalization error, but with a complexity terms of the  KLD  between posterior and prior, which is less tight than the bounds in the paper.  We compared the results of  such an objective ( which is referred to as LAP-KLD in section 6) and showed that it performed much worse.
>
> * We rewrote the proof - hopefully it is clearer.  (see section 3.2 for overview and 8.1 for the full proof).
> In section 8.2  we also added a bound in which the the KL term can vanish at a rate of 1/m (number of samples) if the empirical error is low. For the number of tasks, n, we preferred to keep the 1/n for simplicity and because this term is less important for the LAP algorithm.
>
> * We added a discussion in the introduction (section 1 - first paragraph) about the distinction from continual learning and from multi-task learning. We hope this clarifies our choice of paper title. There is a clear difference from multi-task learning, since the goal in our work is to acquire knowledge (prior) that, when transferred to new tasks, facilitates learning with low generalization error, rather than using multiple tasks collaboratively to aid each task in the given set of tasks.

---

### Official Review · AnonReviewer2 · 2017-11-27
**Interesting algorithm based on a theoretical study, but the main theorem might contain a flaw**

**Rating:** 6
**Confidence:** 4

**Review:**

I personally warmly welcome any theoretically grounded methods to perform deep learning. I read the paper with interest, but I have two concerns about the main theoretical result (Theorem 1, lifelong learning PAC-Bayes bound).
* Firstly, the bound is valid for a [0,1]-valued loss, which does not comply with the losses used in the experiments (Euclidean distance and cross-entropy). This is not a big issue, as I accept that the authors are mainly interested in the learning strategy promoted by the bound. However, this should clearly appear in the theorem statement.
* Secondly, and more importantly, I doubt that the uaw of the meta-posterior as a distribution over priors for each task is valid. In Proposition 1 (the classical single-task PAC-Bayes bound), the bound is valid with probability 1-delta for one specific choice of prior P, and this choice must be independent of the learning sample S. However, it appears that the bound should be valid uniformly for all P in order to be used in Theorem 1 proof (see Equation 18). From a learning point of view, it seems counterintuitive that the prior used in the KL term to learn from a task relies on the training samples (i.e., the same training samples are used to learn the meta-posterior over priors, and the task specific posterior).

A note about the experiments:
I am slightly disappointed that the authors compared their algorithm solely with methods learning from fewer tasks. I would like to see the results obtained by another method using five tasks. A simple idea would be to learn a network independently for each of the five tasks, and consider as a meta-prior an isotropic Gaussian distribution centered on the mean of the five learned weight vectors.

Typos and minor comments:
- Equation 1: \ell is never explicitly defined.
- Equation 4: Please explicitly define m in this context (size of the learning sample drawn from tau).
- Page 4, before Equation 5: A dot is missing between Q and "This".
- Page 7, line 3: Missing parentheses around equation number 12.
- Section 5.1.1, line 5: "The hypothesis class is a the set of..."
- Equation 17: Q_1, ... Q_n are irrelevant.

=== UPDATE ===
I increased my score after author's rebuttal. See my other post.

---

> ### Author Response · Authors · 2017-12-25
> **We thank the reviewer for the helpful comments, and address the  specific points below.**
>
> * We added a comment about the bounded loss issue   (see end of section 2.2).  Indeed,  this is not a big issue since - theoretically, we can claim to bound a truncated version of the loss, and empirically the losses are almost always smaller than one.
>
> * Thank you for pointing out the delicate issue about our main Theorem. We have rewritten the proofs using a different technique, which clarifies the points made by the reviewer and, in fact, leads to improved bounds (see section 3.2 for overview and 8.1 for full proof).
> In the new formulation, each task bound holds for all hyper-posteriors and all posteriors, so it is valid to optimize both using the same samples.
> Note that our new theorem deviates significantly in both proof technique and behavior from that in Pentina and Lampert’s work.
>
> * In section 6, we added the experiment you suggested and several other methods  which use all the training tasks, including:
> 1. Using the bound from  Pentina and Lampert’s work as a learning objective,
> 2. Using an objective derived from variational methods and hierarchical generative models and 3. A recent  method - Model-Agnostic Meta-Learning (MAML, Finn, Abbeel, and Levine. "Model-Agnostic Meta-Learning for Fast Adaptation of Deep Networks." arXiv preprint arXiv:1703.03400 (2017).) .

---

> ### Comment · AnonReviewer2 · 2018-01-12
> **An improved paper**
>
> The authors performed a substantial amount of work to address reviewer comments, both from a theoretical and empirical perspective. The submitted revision turns out to be an improved paper, and I raised my score from 5 to 6.
> In particular, the new PAC-Bayes theorem is much more interesting.
>
> Note that it took me a while to get convinced of the validity of the new proof; I was confused by the fact that the hyper-posterior $\mathcal Q$ relies on the samples S_1, ..., S_i, ..., S_n, whereas this is never explicitly said in the proof of Section 8.1 (see Equation 18). But it turns out that the result is not affected by this. I think this should be made clearer for the readers benefit.
>
> However, the latter point made me realize that the learning algorithm promoted by the theoretical result needs to learn from all tasks simultaneously (it is indeed what is performed in the paper). Considering this, I agree with the two other reviewers that the term "lifelong learning" should not be used here, as there is no continuous learning involved. Personally, I consider this framework as a variant of transfer learning, where one observes multiple tasks before learning a target one. That being said, I conceive that this "overuse" of the buzzword "lifelong learning" has been present in several works lately.

---

> > ### Author Response · Authors · 2018-01-16
> > **Authors response**
> >
> > We appreciate your thorough review which greatly contributes to our work.
> > Regarding the name 'lifelong', we followed the definition of Pentina & Lampert.
> > However, we agree that the name might be misleading and we will change it to 'meta-learning' in future submissions.

---

### Official Review · AnonReviewer3 · 2017-12-04
**Interesting risk bound but empirical evaluation is not convincing**

**Rating:** 6
**Confidence:** 4

**Review:**

The paper considers multi-task setting of machine learning. The first contribution of the paper is a novel PAC-Bayesian risk bound. This risk bound serves as an objective function for multi-task machine learning. A second contribution is an algorithm, called LAP, for minimizing a simplified version of this objective function. LAP algorithm uses several training tasks to learn a prior distribution P over hypothesis space. This prior distribution P is then used to find a posterior distribution Q that minimizes the same objective function over the test task. The third contribution is an empirical evaluation of LAP over toy dataset of two clusters and over MNIST.

While the paper has the title of "life-long learning", the authors admit that all experiments are in multi-task setting, where
the training is done over all tasks simultaneously. The novel risk bound and LAP algorithm can definitely be applied to life-long setting, where training tasks are available sequentially. But since there is no empirical evaluation in this setting, I suggest to adjust the title of the paper.

The novel risk bound of the paper is an extension of the bound from [Pentina & Lampert, ICML 2014]. The extension seems to be quite significant. Unlike the bound of [Pentina & Lampert, ICML 2014], the new bound allows to re-use many different PAC-Bayesian complexity terms that were published previously.

I liked risk bound and optimization sections of the paper. But I was less convinced by the empirical experiments. Since
the paper improves the risk bound of [Pentina & Lampert, ICML 2014], I expected to see an empirical comparison of LAP and optimization  algorithm from the latter paper. To make such comparison fair, both optimization algorithms should use the same base algorithm, e.g. ridge regression, as in [Pentina & Lampert, ICML 2014]. Also I suggest to use the datasets from the latter paper.

The experiment with multi-task learning over MNIST dataset looks interesting, but it is still a toy experiment. This experiment  will be more convincing with more sophisticated datasets (CIFAR-10, ImageNet) and architectures (e.g. Inception-V4, ResNet).

Minor remarks:
Section 6, line 4: "Combing" -> "Combining"
Page 14, first equation: There should be "=" before the second expectation.

---

> ### Author Response · Authors · 2017-12-25
> **We thank the reviewer for the helpful comments, and address the  specific points below.**
>
> * We added a discussion in the introduction about the distinction from multi-task learning  (section 1 - first paragraph).
> There is a clear difference from multi-task, since in lifelong learning  the goal is to acquire knowledge (prior) that when transferred to new tasks facilitates good learning.
> While we call this transfer  setup “lifelong learning ” (as in Pentina and Lampert’s work), it can also be called “learning-to-learn”. But ‘multi-task learning’ is inappropriate because of the different goals and outcome of learning (a prior for learning tasks vs. solutions to given tasks).
>
> * We added an experimental  comparison to a learning objective which is based on Pentina and Lampert’s main theorem.  As can be seen in section 6, this bound  leads to far worse empirical results. We believe that  using our theorem leads to better performance since it is a tighter bound.
>
> * Due to  technical difficulties and lack of time we cannot provide a high quality multiple data-set evaluation at this time.
> However, we did add a comparison to competitive recent  approach - Model-Agnostic Meta-Learning (MAML, Finn, Abbeel, and Levine. "Model-Agnostic Meta-Learning for Fast Adaptation of Deep Networks." arXiv preprint arXiv:1703.03400 (2017).)   (see section 6).

---

> > ### Comment · AnonReviewer3 · 2018-01-23
> > **Authors response**
> >
> > The authors addressed most of my concerns. I will upgrade my score. The only remaining issue is evaluation with more sophisticated datasets and architectures.

---

### Comment · Area_Chair · 2018-01-24
**Dear Authors...**

There are comments by AnonReviewer1 that require your immediate attention and may materially impact your article's acceptance.  Please respond as soon as possible.

Note that OpenReview seems to not be sending email announcements for messages not marked Everyone, so please use that designation.

---

> ### Author Response · Authors · 2018-01-24
> **Response to Area Chair**
>
> We thank the area chair for the helpful comment.
> Indeed there was a problem with the constant, please see our response to AnonReviewer1.
>
> P.S.
> Since the submission of the revised paper we added more experiments that demonstrate the meta-learning  performance in varied task-environments and with different number of training-tasks.
> We would be happy to include those in a revised version if possible.

---

### Decision · Program_Chairs · 2018-01-29
**ICLR 2018 Conference Acceptance Decision**

**Decision:**

Reject

**Comment:**

The author's revisions addressed clarity issues and some experimental issues (e.g., including MAML results in the comparison). The work takes an original path to an important problem (transfer learning, essentially). There is a question of significance, and this is due to the fact that the empirical comparisons are still very limited. The task is an artificial one derived from MNIST. I would call this "toy" as well. On this toy task, the approach isn't that much different from MAML, which is not in of itself a problem, but it would be interested to have a less superficial discussion of the differences.

The authors mention that they didn't have time for a larger empirical study. I think one is necessary in this case because the work is purposing a new learning algorithm/framework, and the question of its potential impact/significance is an empirical one.